# PPAR Pan Agonist MHY2013 Alleviates Renal Fibrosis in a Mouse Model by Reducing Fibroblast Activation and Epithelial Inflammation

**DOI:** 10.3390/ijms24054882

**Published:** 2023-03-02

**Authors:** Minjung Son, Ga Young Kim, Yejin Yang, Sugyeong Ha, Jeongwon Kim, Doyeon Kim, Hae Young Chung, Hyung Ryong Moon, Ki Wung Chung

**Affiliations:** 1Department of Pharmacy, College of Pharmacy, Pusan National University, Busan 46241, Republic of Korea; 2Department of Manufacturing Pharmacy, College of Pharmacy, Pusan National University, Busan 46241, Republic of Korea

**Keywords:** kidney fibrosis, PPAR, inflammation, fibroblasts

## Abstract

The peroxisome proliferator-activated receptor (PPAR) nuclear receptor has been an interesting target for the treatment of chronic diseases. Although the efficacy of PPAR pan agonists in several metabolic diseases has been well studied, the effect of PPAR pan agonists on kidney fibrosis development has not been demonstrated. To evaluate the effect of the PPAR pan agonist MHY2013, a folic acid (FA)-induced in vivo kidney fibrosis model was used. MHY2013 treatment significantly controlled decline in kidney function, tubule dilation, and FA-induced kidney damage. The extent of fibrosis determined using biochemical and histological methods showed that MHY2013 effectively blocked the development of fibrosis. Pro-inflammatory responses, including cytokine and chemokine expression, inflammatory cell infiltration, and NF-κB activation, were all reduced with MHY2013 treatment. To demonstrate the anti-fibrotic and anti-inflammatory mechanisms of MHY2013, in vitro studies were conducted using NRK49F kidney fibroblasts and NRK52E kidney epithelial cells. In the NRK49F kidney fibroblasts, MHY2013 treatment significantly reduced TGF-β-induced fibroblast activation. The gene and protein expressions of collagen I and α-smooth muscle actin were significantly reduced with MHY2013 treatment. Using PPAR transfection, we found that PPARγ played a major role in blocking fibroblast activation. In addition, MHY2013 significantly reduced LPS-induced NF-κB activation and chemokine expression mainly through PPARβ activation. Taken together, our results suggest that administration of the PPAR pan agonist effectively prevented renal fibrosis in both in vitro and in vivo models of kidney fibrosis, implicating the therapeutic potential of PPAR agonists against chronic kidney diseases.

## 1. Introduction

Chronic kidney disease (CKD) affects approximately 10% of the global population, with high mortality due to limited treatment options [1]. CKD often leads to end-stage renal disease, which is fatal without renal replacement therapy, such as dialysis or kidney transplantation. Kidney fibrosis is considered a major underlying pathological process that is commonly detected in CKD development [2]. Understanding the mechanisms of renal fibrosis is essential for developing therapies to prevent or slow CKD progression. Fibrosis is defined by the formation and accumulation of the extracellular matrix (ECM), mainly by tissue-resident fibroblast cells [3]. Under physiological conditions, minimal amounts of ECM support kidney structure and function. In response to tissue injury, wound-healing processes are activated to inhibit the inflammatory response with proper tissue regeneration. However, persistent inflammatory responses result in incomplete regeneration, with the formation of fibrotic scar tissue [4]. Exaggerated deposition of ECM during chronic and pathological fibrosis development disrupts the normal kidney architecture and interferes with kidney function. At a certain stage, unresolved kidney fibrosis becomes irreversible and contributes to renal failure.

The mechanisms underlying the development of kidney fibrosis have been studied extensively [5]. Regardless of the trigger, multiple cell types participate in fibrogenesis, including fibroblasts, pericytes, epithelial cells, endothelial cells, and inflammatory cells [6]. The main contributor to fibrosis progression is the accumulation of fibroblasts with a phenotypic appearance of myofibroblasts. During progressive fibrosis, the interstitium is filled with myofibroblasts, which produce large amounts of ECM proteins [6]. Although myofibroblasts are the executing cells of fibrosis, other cells also contribute to the development of fibrosis through both direct and indirect mechanisms. Pericytes, epithelial cells, and endothelial cells have been shown to directly contribute to fibrosis through the transition to mesenchymal-like cell types [7]. Epithelial cells also contribute to fibrosis through the secretion of pro-fibrogenic and pro-inflammatory factors, such as TGF-β, CTGF, and cytokines [8,9]. Considerable evidence suggests that inflammatory cells play a critical role in the initiation and progression of renal fibrosis [10,11]. The chemokine is mainly secreted from tubule epithelial cells during injury and recruits various inflammatory cell types, including monocytes, T cells, dendritic cells, and fibrocytes [9]. The infiltration of inflammatory cells is a major phenotype of kidney fibrosis that promotes fibrosis [12].

Peroxisome proliferator-activated receptors (PPARs), PPARα, PPARβ/δ, and PPARγ, play an essential role in the regulation of various physiological processes, including lipid and energy metabolism [13]. Fibrates (PPARα agonists) are used to treat dyslipidemia, and thiazolidinediones (PPARγ agonists) are used to increase insulin sensitivity in type 2 diabetics. In addition, PPAR dual agonists have been developed to treat type 2 diabetes with secondary cardiovascular complications [14,15]. Many synthetic ligands for PPARs are still under development to expand their therapeutic applications. In addition to their original roles in metabolism, PPAR agonists have been shown to exert various physiological effects. PPAR agonists have been reported to block the development of fibrosis in the liver, heart, kidneys, and lungs [16,17]. Furthermore, several studies have reported the anti-inflammatory action of peroxisome proliferator-activated receptor (PPAR) agonists [18].

Previously, we synthesized and evaluated the role of MHY2013, a potent PPAR pan-agonist, in several metabolic disease models [19,20]. In addition, MHY2013 showed anti-fibrotic effects in an age-related renal fibrosis model by regulating the lipid metabolism in epithelial cells [21]. However, the effects of MHY2013 on general aspects of renal fibrosis have not yet been investigated. In this study, we demonstrated the role and efficacy of MHY2013 in a general renal fibrosis model. Using mouse models of renal fibrosis induced by folic acid, we demonstrated the anti-fibrotic efficacy of PPAR pan agonism in renal fibrosis. MHY2013 treatment significantly reduced fibrosis and inflammation in a mouse model of renal fibrosis. In addition, using in vitro analysis, we found anti-fibrotic and anti-inflammatory effects of MHY2013 in renal fibroblasts and epithelial cells.

## 2. Results

### 2.1. MHY2013 Reduces Folic-Acid-Induced Renal Damage and Tubule Dilation in Mice

To evaluate the anti-fibrotic effects of MHY2013, folic-acid-induced renal fibrosis models were used. MHY2013 was intraperitoneally administered at a low (0.5 mg/kg/day) or high dose (3 mg/kg/day) during the experimental period (Figure 1A). The MHY2013-treated group showed lower expression of kidney damage-related genes (*Havcr1*, *Timp2*, *Igfbp7*, and *Spp1*) than those of the FA-treated group (Figure 1B). Blood urea nitrogen (BUN) levels were increased in the folic acid (FA)-treated group, and high-dose MHY2013 treatment significantly blocked the FA-induced BUN increase (Figure 1C). Structural changes were analyzed with hematoxylin and eosin (H&E) staining. Tubule dilation and damage were detected in the cortex and medulla regions of FA-treated kidneys (Figure 1D). MHY2013-treated kidneys showed a smaller increase in tubule dilation (Figure 1D). These results indicate that MHY2013 has protective effects against folic-acid-induced kidney damage.

### 2.2. MHY2013 Suppresses FA-Induced Renal Fibrosis Development in Mice

We further analyzed the effects of MHY2013 on the development of renal fibrosis. The MHY2013-treated group showed lower expression of fibrosis-related genes (*Col1a2*, *Col3a1*, *Vim*) than that of the FA-treated group (Figure 2A). The increased expression of *Col1a2* and *Vim* was confirmed with in situ hybridization (ISH) analysis. FA treatment significantly increased *Col1a2* and *Vim* expression in the interstitial region of the kidney, and MHY2013-treated groups showed lower *Col1a2* and *Vim* expression (Figure 2B,C). The protein levels of fibrosis markers were further checked. FA-induced α-SMA and collagen I levels were significantly decreased with MHY2013 treatment (Figure 3A). An immunohistochemical analysis confirmed that fewer αSMA-positive myofibroblasts were detected in the MHY2013-treated kidneys (Figure 3B). The extent of fibrosis was confirmed using Sirius Red (SR) staining. The FA treatment significantly increased SR-positive regions, whereas the MHY2013 treatment reduced SR-positive regions (Figure 3C,D). Finally, the activation of SMAD proteins was detected. Less SMAD2 and SMAD3 phosphorylation was detected in the MHY2013-treated groups than in the FA groups (Figure 3E). Collectively, these data indicate that MHY2013 effectively blocked FA-induced kidney fibrosis.

### 2.3. FA-Induced Inflammatory Responses Are Down-Regulated by MHY2013

The development of fibrosis is accompanied by pro-inflammatory responses. FA treatment also increases the inflammatory responses in the kidneys [22]. We further examined the inflammatory responses in animal models. MHY2013 treatment significantly reduced pro-inflammatory gene (*Tnfa*, *Il1b*, and *Ccl2*) expression and the macrophage marker *Emr1* in the kidneys (Figure 4A). The activation of NF-κB, induced by FA, was effectively blocked with MHY2013 treatment (Figure 4B). Activated NF-κB was mainly detected in the epithelial cells of dilated tubules, and MHY2013 significantly reduced p-NF-κB expression in tubule cells (Figure 4C). Macrophage infiltration was confirmed using ISH analysis. Increased *Emr1* expression was mainly detected in the interstitial region of FA-treated kidneys (Figure 4D). MHY2013-treated groups showed less macrophage infiltration in the kidneys (Figure 4D). We further detected the co-expression of *Col1a2* and *Emr1*. In the FA group, *Emr1*- and *Col1a2*-positive cells were colocalized in the kidneys, indicating that inflammation is connected to fibrosis development (Figure 4D). In accordance with the qPCR results, the MHY2013-treated group showed lower *Emr1* and *Col1a2* expression in the kidney (Figure 4D). These results indicate that MHY2013 exerts anti-inflammatory effects against FA-induced kidney fibrosis.

### 2.4. MHY2013 Blocks TGF-β-Induced NRK49F Kidney Fibroblast Activation

To investigate the anti-fibrotic role of MHY2013 under in vitro conditions, we used kidney-derived fibroblast cells. First, we confirmed the activation of PPAR by MHY2013 in NRK49F kidney fibroblasts. MHY2013 significantly increased PPRE activity under PPARα, PPARβ, and PPARγ expression conditions, confirming MHY2013 as a PPAR pan agonist (Figure 5A–C). TGF-β treatment significantly increased *Col1a2*, *Acta2*, and *Vim* expression in NRK49F fibroblasts, and MHY2013 pre-treatment effectively blocked fibroblast activation (Figure 5D). The protein expression levels of α-SMA and Col1 were analyzed. MHY2013 treatment significantly reduced TGF-β-induced α-SMA and Col1 protein expression (Figure 5E). The increased expression of α-SMA was confirmed using immunofluorescence. TGF-β increased αSMA expression in cells, whereas MHY2013 reduced αSMA expression (Figure 5F). To examine which PPAR subtype influenced fibroblast activation, we overexpressed PPAR before TGF-β treatment. We found that PPARγ overexpression effectively blocked TGF-β-induced fibroblast activation (Figure 5G), whereas other PPAR subtypes did not show a significant reduction (data not shown). These results indicate that MHY2013 effectively blocks TGF-β-induced NRK49F kidney fibroblast activation, mainly through PPARγ activation.

### 2.5. MHY2013 Reduces LPS-Induced Chemokine Expression in NRK52E Kidney Epithelial Cells

To examine the anti-inflammatory effects of MHY2013 under in vitro conditions, kidney tubule epithelial cells were used. Stimulation of NRK52E cells with a lipopolysaccharide (LPS) significantly increased chemokine gene expression, and MHY2013 pretreatment effectively reduced their expression (Figure 6A). We further evaluated NF-κB activity using a luciferase assay. LPS treatment significantly increased NF-κB activity, whereas MHY2013 effectively blocked NF-κB activity (Figure 6B). Finally, to examine which PPAR subtype influences LPS-induced chemokine expression, we overexpressed PPAR before LPS treatment. We found that PPARβ overexpression effectively blocked LPS-induced chemokine expression (Figure 6C), whereas other PPAR subtypes did not show a significant reduction (data not shown). Collectively, these data show that MHY2013 reduces LPS-induced NF-κB activation and chemokine expression in renal epithelial cells, mainly through PPARβ activation.

## 3. Discussion

Renal fibrosis, which is generally accompanied by CKD progression, is defined by the loss of renal parenchymal cells and their substitution with ECM proteins. During fibrosis development, both the synthesis and degradation of ECM proteins occur via several intra- and extracellular events. When ECM protein synthesis exceeds degradation, excessive ECM accumulation results in fibrosis [23]. It is well established that various cell types directly and indirectly participate in fibrosis development. Resident fibroblasts are the main responsible cells for the synthesis of ECM proteins [3]. During fibrogenesis, fibroblasts receive signals from other cells and begin to proliferate and become myofibroblasts. Myofibroblasts produce large amounts of ECM proteins that primarily contribute to the pathogenesis of kidney fibrosis.

Transforming growth factor-β (TGF-β) is considered a key player of renal fibrosis by stimulating fibroblasts in the kidney, thus making it an interesting target for the treatment of fibrosis [24]. Indeed, anti-TGF-β treatments using neutralizing antibodies, inhibitors against the TGF-β receptor, or antisense oligonucleotides to TGF-β1 halt the progression of renal fibrosis development, suggesting its fibrotic role in CKD [25]. We found that MHY2013 significantly reduced TGF-β-induced fibroblast activation in vitro. MHY2013 effectively inhibits TGF-β-induced α-SMA and collagen I expression in fibroblasts. Several studies have reported that PPARγ activation blocks TGF-β-induced ECM production in fibroblasts. Wang et al. evaluated three PPARγ agonists (15d-PGJ2, troglitazone, and ciglitazone) and found that PPARγ activation directly inhibits TGF-β/SMAD signaling pathways and alleviates renal fibroblast activation, resulting in reduced ECM synthesis [26]. Another PPARγ agonist, pioglitazone, similarly prevents renal fibrosis by repressing the TGF-β signaling pathway [27]. MHY2013 also showed direct anti-fibrotic effects on fibroblasts. Using PPAR transfection, we found that PPARγ overexpression inhibits TGF-β-induced fibroblast activation. Based on these results, we concluded that MHY2013 directly reduces fibroblast activation through PPARγ activation.

Renal inflammation is a protective response induced during kidney injury, which eliminates the cause of injury and promotes tissue repair. However, unresolved inflammatory responses can promote abnormal fibrosis in the kidneys, leading to CKD [28]. During prolonged inflammation, bone-marrow-derived leukocytes, including neutrophils and macrophages, are the main players in kidney inflammation. The accumulation of these cells is a major feature of pro-inflammatory kidney disease. In addition to these cells, studies have also revealed the important role of locally activated kidney cells, such as tubular epithelial cells (TECs), mesangial cells, podocytes, and endothelial cells. During the development of interstitial fibrosis, TECs play an important role in initiating the inflammatory response [29]. Under damaged conditions, TECs actively participate in pro-inflammatory responses through chemokine production. Several lines of evidence suggest that chemokines produced from TECs are crucial for the recruitment of monocytes and macrophages [30]. Based on these observations, the regulation of epithelial inflammation has been an interesting target for modulating kidney inflammation and fibrosis.

Based on our finding that MHY2013 decreases inflammation in animal models, we further demonstrated its role in epithelial inflammation. MHY2013 significantly reduces NF-κB activation and chemokine production in epithelial cells. Furthermore, using PPAR subtype transfection, we found that PPARβ overexpression decreases chemokine production in epithelial cells. There is evidence that PPARβ exerts anti-inflammatory effects in kidney disease. PPARβ-null mice developed more severe ischemic renal injury with more severe tubule damage than wild-type mice [31]. A macrophage-specific PPARβ-deleted mouse model also showed impaired apoptotic cell clearance and reduced anti-inflammatory cytokine production [32]. These mice were much more likely to develop autoimmune kidney disease, a lupus-like autoimmune disease. In addition, several reports have demonstrated the anti-inflammatory role of PPARβ agonists in kidney disease. GW0742 has been shown to inhibit streptozotocin-induced diabetic nephropathy in mice by reducing inflammatory mediators, including MCP-1 and osteopontin [33]. Another study showed that PPARβ agonists reduced the incidence of hypertension, endothelial dysfunction, inflammation, and organ damage in lupus mice [34]. Collectively, the reduced inflammatory responses observed in our in vitro and in vivo experiments were associated with the PPARβ activation property of MHY2013.

## 4. Materials and Methods

### 4.1. Animal Studies

All animal experiments were approved by the Institutional Animal Care Committee of the Pusan National University (PNU-IACUC approval No. PNU-2022-3164) and performed according to the guidelines issued by Pusan National University. C57BL/6J mice were obtained from Hyochang Science (Daegu, Republic of Korea). To establish the renal fibrosis mouse model, male mice (7-week-old) were intraperitoneally injected with a single dose of folic acid (250 mg/kg dissolved in 0.3 M NaHCO_3_) or vehicle. For the MHY treatment groups, MHY2013 was intraperitoneally administered in low (0.5 mg/kg/day) or high doses (3 mg/kg/day) during the experimental period (*n* = 5~7). All mice were maintained at 23 ± 2 °C with a relative humidity of 60 ± 5% and 12 h light/dark cycles. One week after the folic acid treatment, the mice were sacrificed using CO_2_ inhalation. Serum was collected for biochemical analyses. Kidneys were collected and then immediately frozen in liquid nitrogen. For long-term storage, kidney samples were moved to a −80 °C deep freezer. Part of kidneys was fixed in neutral-buffered formalin for histochemical experiments.

### 4.2. Cell Culture Experiments

NRK49F rat-kidney fibroblasts were purchased from ATCC (CRL-1570) and grown in Dulbecco’s modified Eagle’s medium (DMEM), supplemented with 10% fetal bovine serum (FBS) and 1% penicillin. All cells were incubated at 5% CO_2_ and 37 °C in a water-saturated atmosphere. To determine the effect of MHY2013 on TGFβ-induced fibroblast activation and ECM production, a MHY2013 concentration with 10 μM was pre-treated 30 min before the TGFβ (10 ng/mL) treatment. Protein or RNA samples were collected 24 h after the TGF-β treatment to determine the effect of MHY2013. NRK52E rat-kidney epithelial cells were purchased from ATCC (CRL-1571) and grown in DMEM supplemented with 10% FBS and 1% penicillin. To determine the effect of MHY2013 on LPS-induced inflammation, a MHY2013 concentration of 10 μM was pre-treated 30 min before LPS (10 μg/mL) treatment. Protein and RNA samples were collected 1 h after LPS treatment to determine the effect of MHY2013. All cell culture experiments were performed at least 3 times per experiment.

### 4.3. Serum Biochemical Measurements

Serum samples were obtained using centrifugation at 3000 rpm for 20 min at 4 °C. Blood urea nitrogen (BUN) levels were measured using a commercial assay kit from Shinyang Diagnostics (SICDIA L-BUN, 1120171, Seoul, Republic of Korea) according to the manufacturer’s instructions.

### 4.4. Protein Extraction and Western Blot Analysis

Two different solutions were used to extract proteins: ProEXTM CETi protein extract solution (Translab, Daejeon, Republic of Korea) was used to extract protein from tissues, and RIPA buffer (#9806, Cell Signaling Technology, Danvers, MA, USA) was used to obtain the total protein from the cells. Both solutions contained protease inhibitor cocktails to prevent protein degradation and phosphate inhibitor to prevent dephosphorylation. Protein concentration was measured using a BCA reagent (Thermo Scientific, Waltham, MA, USA). Extracted proteins (5–20 μg of protein) were then mixed with 4× sample buffer (Cat#1610747, Bio-Rad, CA, USA) and boiled for 5 min. The proteins were then separated using sodium dodecyl sulfate-polyacrylamide gel electrophoresis and transferred to polyvinylidene difluoride membranes (Millipore, Burlington, MA, USA). The membranes were blocked in 5% nonfat milk and washed with Tris-buffered saline-Tween buffer for 30 min. Specific primary antibodies (1:500 to 1:2000 dilution, Appendix A) were added to the membranes and incubated overnight at 4 °C. After three washes with the TBS-Tween buffer, the membranes were incubated with a horseradish peroxidase-conjugated anti-mouse, anti-rabbit, or anti-goat antibody (diluted 1:10,000) for 1 h at 25 °C. The resulting immunoblots were visualized using Western Bright Peroxide solution (Advansta, San Jose, CA, USA) and a ChemiDoc imaging system (Bio-Rad) according to the manufacturer’s instructions. All western blot analyses were performed at least 3 times per experiment.

### 4.5. RNA Extraction and qRT-PCR

Total RNA was prepared using a TRIzol reagent (Invitrogen, Carlsbad, CA, USA). Briefly, kidney tissues (*n* = 5~7) or cells (*n* = 3) were homogenized in the TRIzol reagent. To isolate RNA, 0.2 mL chloroform was added to the 1 mL homogenate and shaken vigorously for 15 min. The aqueous phases were transferred to fresh tubes, and an equal volume of isopropanol was added. The samples were then incubated at 4 °C for 15 min and centrifuged at 12,000× *g* for 15 min at 4 °C. The supernatants were removed, and the resulting RNA pellets were washed once with 75% ethanol and then dried, followed by dissolving in diethyl pyrocarbonate-treated water. Next, 1.0 μg of isolated RNA was reverse-transcribed using a cDNA synthesis kit from GenDEPOT (Katy, TX, USA). qPCR was performed using a SYBR Green Master Mix (BIOLINE, Taunton, MA, USA) and a CFX Connect System (Bio-Rad). Primers were designed using Primer3Plus [35], and the primer sequences used are listed in Appendix A. For qPCR data analysis, the 2^−ΔΔCT^ method was used as a relative quantification strategy.

### 4.6. Histological Analysis

To visualize histological changes in the kidneys, the kidneys were fixed in 10% neutral formalin, and paraffin-embedded sections were stained with H&E. To assess the degree of renal fibrosis and damage, SR staining was performed using a commercially available kit (VB-3017; Rockville, MD, USA). This staining method is commonly used to visualize collagen fibers, which are a hallmark of fibrosis. Immunohistochemical analysis was performed to visualize the protein expression regions in the kidneys. Briefly, paraffin-embedded sections were deparaffinized and rehydrated. The sections were then incubated with the primary antibodies and visualized using diaminobenzidine substrates. The sections were counterstained with hematoxylin, which allows for the visualization of cell nuclei. Images were obtained using a microscope (LS30; Leam Solution, Seoul, Republic of Korea).

### 4.7. In Situ Hybridization

ISH was performed using formalin-fixed paraffin-embedded tissue samples. RNAscope 2.5 HD Assay (322300, Biotechne, Minneapolis, MN, USA) or RNAscope 2.5 HD Duplex Detection Kit (322436, bio-techne, Minneapolis, MN, USA) was used to visualize RNA expression in the tissue, in accordance with the manufacturer’s instructions. The following probes were used to perform the RNAscope assay: Mm-Vim cat# 457961, Mm-Emr1 cat# 317969-C2, and Mm-Col1a1 cat# 319379. Images were obtained using a microscope (LS30; LEAM Solution, Seoul, Republic of Korea).

### 4.8. Measurement of Transcriptional Activity

Luciferase assays were performed to determine the transcriptional activity of PPAR transcription factors in the NRK49F cells. Briefly, NRK49F cells were transfected with the PPRE-X3-TK-LUC plasmid (0.1 µg) with PPARα, PPARβ/δ, or PPARγ expression vectors (0.01 µg) using Lipofectamine 3000 reagent (Invitrogen, Carlsbad, CA, USA.). The cells were further treated with MHY2013 or WY14643 (a known PPARα agonist), GW501516 (a known PPARβ/δ agonist), and rosiglitazone (a known PPARγ agonist). The luciferase activity was measured using a One-Glo Luciferase Assay System (Promega, Madison, WI, USA). After adding the luciferase substrate, the luminescence was measured using a luminescence plate reader (Berthold Technologies GmbH & Co., Bad Wildbad, Germany). Luciferase assays were performed to determine the transcriptional activity of NF-κB in the NRK52E cells. The cells were transfected with the NF-κB promoter-LUC plasmid, and the luciferase activity was measured using a One-Glo Luciferase Assay System and a luminescence plate reader.

### 4.9. Immunofluorescence

Immunofluorescence was performed to visualize protein expression in the cells. The cells were fixed in 4% formaldehyde for 10 min, washed thrice with ice-cold PBS, and exposed to 0.25% Triton-X 100 in PBS for 10 min for permeabilization. To prevent non-specific binding of antibodies, the cells were blocked using a solution containing 1% BSA and 0.1% Tween 20 in PBS at room temperature for 30 min. Next, the cells were incubated overnight with anti-αSMA antibody, which had been diluted in the blocking buffer at 4 °C. After washing off any unbound antibodies with PBS, the cells were incubated with a secondary antibody conjugated with a fluorescent tag for 1 h in the dark. The cells counterstained with Hoechst 33258 in PBS for 1 min to visualize the nuclei. The images were captured using a fluorescence microscope (LS30).

### 4.10. Quantification and Statistical Analysis

Student’s *t*-test was used to analyze the differences between the two groups, and an analysis of variance was used to analyze intergroup differences. The level of statistical significance was set at *p* < 0.05. The software used for the analyses was GraphPad Prism version 5 (GraphPad Software Inc., San Diego, CA, USA). Image calculations were performed using the ImageJ software (National Institutes of Health, Bethesda, MD, USA).

## 5. Conclusions

In conclusion, we investigated the anti-fibrotic and anti-inflammatory roles of the PPAR pan agonist MHY2013 using in vitro and in vivo kidney fibrosis models. When administered to the FA-induced mouse kidney fibrosis model, MHY2013 effectively reduced fibrosis development and inflammatory responses in the kidney. The anti-fibrotic and anti-inflammatory mechanisms of MHY2013 were further demonstrated using NRK49F kidney fibroblasts and NRK52E kidney epithelial cells. MHY2013 directly reduced TGF-β-induced ECM production in fibroblasts mainly through PPARγ activation, whereas MHY2013 suppressed LPS-induced pro-inflammatory responses in TECs mainly through PPARβ activation. Taken together, our results suggest that the administration of the PPAR pan agonist effectively prevented renal fibrosis in both in vitro and in vivo models of kidney fibrosis, implicating the therapeutic potential of PPAR agonists against chronic kidney diseases (Figure 7).

## Figures and Tables

**Figure 1 ijms-24-04882-f001:**
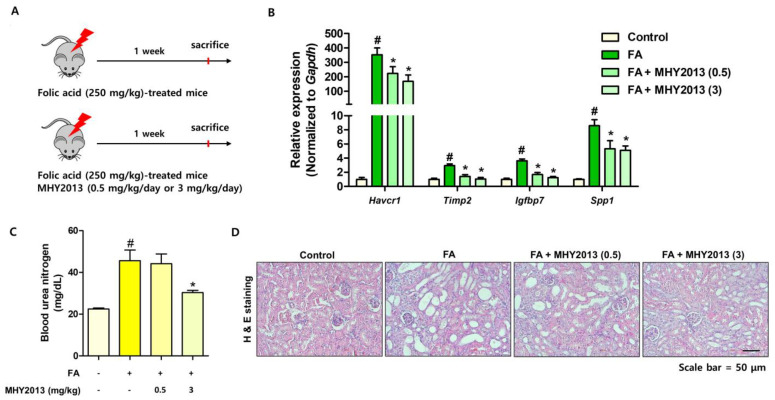
MHY2013 treatment reduces kidney damage induced with folic acid (FA) treatment. (**A**) Experimental scheme to evaluate the effect of MHY2013 using FA-induced mouse kidney fibrosis model (*n* = 5~7). (**B**) Relative mRNA levels of kidney damage-related genes (*Havcr1*, *Timp2*, *Igfbp*, and *Spp1*) in FA-treated kidneys with or without MHY2013 treatment. # *p* < 0.05 vs. control groups. * *p* < 0.05 vs. FA group. (**C**) Blood urea nitrogen (BUN) levels in FA-treated serums with or without MHY2013. # *p* < 0.05 vs. control groups. * *p* < 0.05 vs. FA group. (**D**) Representative images of H&E-stained sections of FA and/or MHY2013-treated kidneys.

**Figure 2 ijms-24-04882-f002:**
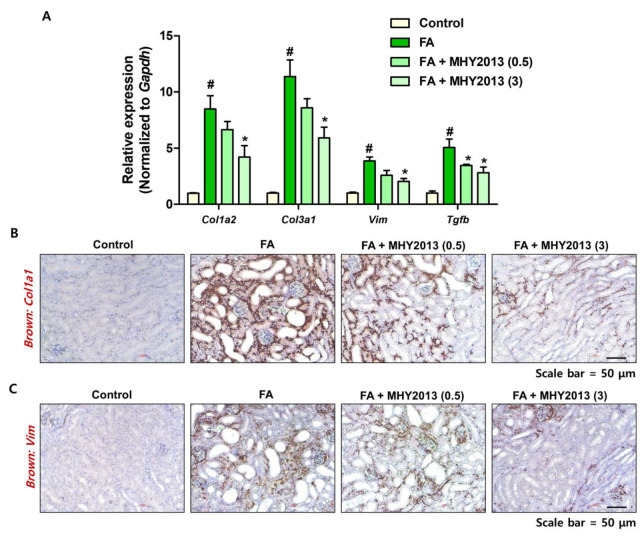
MHY2013 treatment reduces fibrosis-related gene expression in FA model. (**A**) Relative gene expression levels of kidney fibrosis-related proteins (*Col1a2*, *Col3a1*, *Vim*, and *Tgfb*) in FA-treated kidneys with or without MHY2013 treatment. # *p* < 0.05 vs. control groups. * *p* < 0.05 vs. FA group. (**B**) Representative in situ hybridization images detected with *Col1a1* (brown) probe in FA-treated kidneys with or without MHY2013 treatment. (**C**) Representative in situ hybridization images detected with *Vim* (brown) probe in FA-treated kidneys with or without MHY2013 treatment.

**Figure 3 ijms-24-04882-f003:**
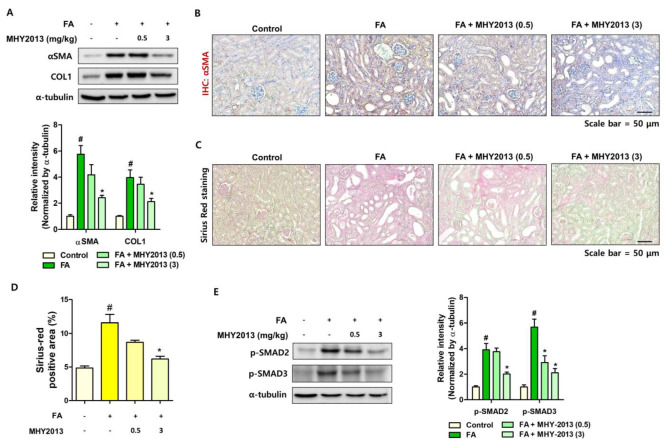
MHY2013 treatment reduces fibrosis-related protein expression in FA model. (**A**) αSMA and COL1 protein levels were determined in FA-treated kidneys with or without MHY2013 treatment. α-tubulin was used as a loading control. Fold change differences in protein expression were determined with densitometric analysis. # *p* < 0.05 vs. control groups. * *p* < 0.05 vs. FA group. (**B**) Representative immunohistochemical (IHC) images of αSMA expression in FA-treated kidneys with or without MHY2013 treatment. (**C**) Representative images of Sirius Red staining in FA-treated kidneys with or without MHY2013 treatment. (**D**) Positive areas for Sirius Red staining were calculated to determine the extent of fibrosis in each group. # *p* < 0.05 vs. control groups. * *p* < 0.05 vs. FA group. (**E**) Protein levels of p-SMAD2 and p-SMAD3 were determined in FA-treated kidneys with or without MHY2013 treatment. α-tubulin was used as a loading control. Fold change differences in protein expression were determined with densitometric analysis. # *p* < 0.05 vs. control groups. * *p* < 0.05 vs. FA group.

**Figure 4 ijms-24-04882-f004:**
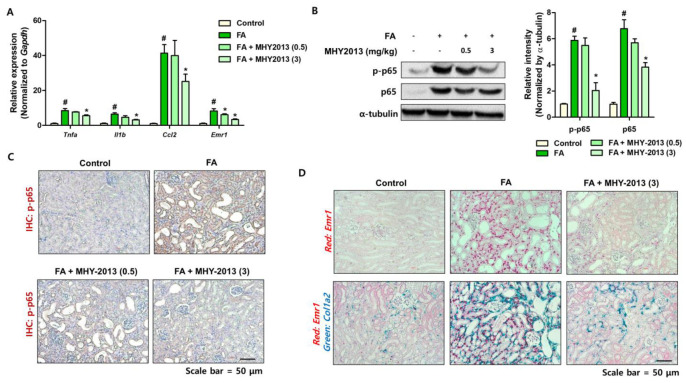
MHY2013 treatment reduces inflammation in FA model. (**A**) Relative mRNA levels of inflammation-related genes (*Tnfa*, *Il1b*, *Ccl2*, and *Emr1*) in FA-treated kidneys with or without MHY2013 treatment. # *p* < 0.05 vs. control groups. * *p* < 0.05 vs. FA group. (**B**) Protein levels of p-p65 and p65 were determined in FA-treated kidneys with or without MHY2013 treatment. α-tubulin was used as a loading control. Fold change difference in protein expression were determined with densitometric analysis. # *p* < 0.05 vs. control groups. * *p* < 0.05 vs. FA group. (**C**) Representative IHC images of p-p65 expression in FA-treated kidneys with or without MHY2013 treatment. (**D**) Representative in situ hybridization images detected with *Emr1* (red) and *Col1a2* (green) probes in FA-treated kidneys with or without MHY2013 treatment.

**Figure 5 ijms-24-04882-f005:**
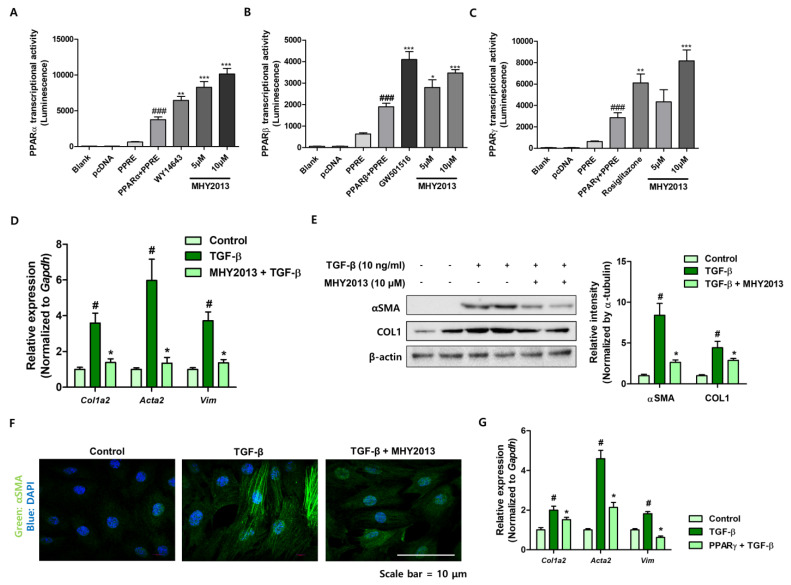
MHY2013 inhibits TGF-β-induced fibroblast activation in NRK49F cells. (**A**) The effect of MHY2013 on PPARα transcriptional activity was measured using the PPRE luciferase system. ### *p* < 0.001 vs. PPRE group. ** *p* < 0.005 vs. PPARα + PPRE group. *** *p* < 0.001 vs. PPARα + PPRE group. (**B**) The effect of MHY2013 on PPARβ transcriptional activity was measured using the PPRE luciferase system. ### *p* < 0.001 vs. PPRE group. * *p* < 0.05 vs. PPARβ + PPRE group. ** *p* < 0.005 vs. PPARβ + PPRE group. *** *p* < 0.001 vs. PPARβ + PPRE group. (**C**) The effect of MHY2013 on PPARγ transcriptional activity was measured using the PPRE luciferase system. ### *p* < 0.001 vs. PPRE group. ** *p* < 0.005 vs. PPARγ + PPRE group. *** *p* < 0.001 vs. PPARγ + PPRE group. (**D**) Relative mRNA levels of kidney fibrosis-related genes (*Col1a2*, *Acta2*, and *Vim*) in TGF-β-treated NRK49F cells with or without MHY2013. # *p* < 0.05 vs. control group. * *p* < 0.05 vs. TGF-β group. (**E**) Protein levels of αSMA and Collagen Ⅰ were determined in TGF-β-treated NRK49F cells with or without MHY2013. β-actin was used as a loading control. Fold change differences in protein expression were determined with densitometric analysis. # *p* < 0.05 vs. control groups. * *p* < 0.05 vs. TGF-β group. (**F**) Representative immunofluorescence images of α-SMA expression (green) in TGF-β-treated NRK49F cells with or without MHY2013. Nuclei were counterstained with DAPI. (**G**) Relative mRNA levels of kidney fibrosis-related genes (*Col1a2*, *Acta2*, and *Vim*) in TGF-β-treated NRK49F cells with or without PPARγ expression.

**Figure 6 ijms-24-04882-f006:**
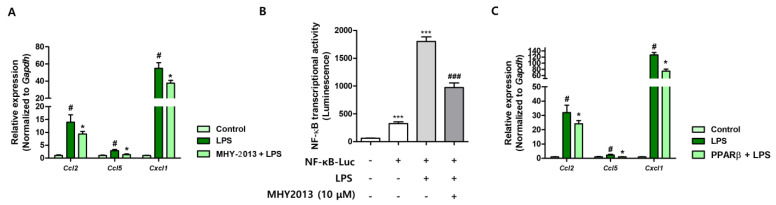
MHY2013 suppresses LPS-induced inflammatory responses in NRK52E cells. (**A**) Relative mRNA levels of inflammation-related genes (*Ccl2*, *Ccl5*, and *Cxcl1*) in LPS-treated NRK52E cells with or without MHY2013. # *p* < 0.05 vs. control group. * *p* < 0.05 vs. LPS group. (**B**) NF-κB transcriptional activity was measured under LPS-treated condition with or without MHY2013 treatment. *** *p* < 0.001 vs. non-treated group. ### *p* < 0.001 vs. LPS-treated group. (**C**) Relative mRNA levels of inflammation-related genes (*Ccl2, Ccl5*, and *Cxcl1*) in LPS-treated NRK52E cells with or without PPARβ expression.

**Figure 7 ijms-24-04882-f007:**
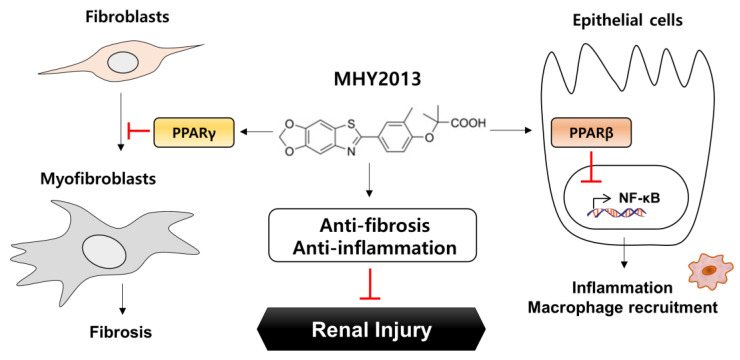
MHY2013 alleviates renal fibrosis in a mouse model by reducing fibroblast activation and epithelial inflammation.

## Data Availability

Data is contained within the article.

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
