# Peer review of "PPAR Pan Agonist MHY2013 Alleviates Renal Fibrosis in a Mouse Model by Reducing Fibroblast Activation and Epithelial Inflammation"

_ijms, 2023, doi:10.3390/ijms24054882_

Round 1
Reviewer 1 Report
The work by Son et al. on PPAR pan agonist MHY-2013 alleviates renal fibrosis in a mouse model by reducing fibroblast activation and epithelial inflammation is an extremely interesting piece of literature that requires several significant changes:
- I would like to ask you to reformat the work to the format corresponding to the journal to which the work was submitted;
- please clearly specify the purpose of the work in the introduction section
- please number the relevant sections 1. Introduction, 2. Result etc.
- please explain what "+" and "-" used in figure 1C, 3AED, 4B, 5E, 6B mean
- please number the subsections in the Result section
- please add a conclusion section, which will summarize all the work
Author Response
The work by Son et al. on PPAR pan agonist MHY-2013 alleviates renal fibrosis in a mouse model by reducing fibroblast activation and epithelial inflammation is an extremely interesting piece of literature that requires several significant changes:
Response to reviewer: Thanks for your helpful comments.
- I would like to ask you to reformat the work to the format corresponding to the journal to which the work was submitted;
Response to reviewer: We reformatted our manuscripts according to the journal style.
- please clearly specify the purpose of the work in the introduction section.
Response to reviewer: We added the purpose of the work at the end of the introduction section.
- please number the relevant sections 1. Introduction, 2. Result etc.
Response to reviewer: We numbered all sections.
- please explain what "+" and "-" used in figure 1C, 3AED, 4B, 5E, 6B mean
Response to reviewer: The “+” means that the things in the right are added and the “-“ means that the things in the right are not added in the experimental condition.
- please number the subsections in the Result section
Response to reviewer: We numbered the subsections in the result section.
- please add a conclusion section, which will summarize all the work
Response to reviewer: We added a conclusion section which summarize all the work.
Reviewer 2 Report
Son et al. report the protective effects of PPAR pan agonist, MHY-2013, on the folic acid-induced kidney fibrosis model. This work suggests that regulating PPAR activity might have protective effects on the kidney fibrosis process, well-designed experimental work. I have minor comments.
1. Figures 5 and 6 suggest that different kinds of PPAR subtypes might have different roles in the kidney fibrosis process. MHY-2013 blocks TGF-β-induced fibroblast activation, mainly through PPARγ activation. However, anti-inflammatory effects have through PPARβ activation. Why did you use different types of cell lines, such as NRK49F cells and NRK52E cells? And why did you use different types of stimulation, such as TGF-β and LPS? Authors need to describe clearly.
2. Subtitle “MHY-2013 reduces LPS-induced chemokine expression in NRK49E kidney epithelial cells“ à NRK49E cells should change to NRK52E cells.
3. NRK49F cells and NRK52E cells might have different cell behaviors. So, in figure 6, the authors need to evaluate PPARα, PPARβ, and PPARγ transcriptional activity using NRK52E cells.
Author Response
Son et al. report the protective effects of PPAR pan agonist, MHY-2013, on the folic acid-induced kidney fibrosis model. This work suggests that regulating PPAR activity might have protective effects on the kidney fibrosis process, well-designed experimental work. I have minor comments.
Response to reviewer: Thanks for your helpful comments.
- Figures 5 and 6 suggest that different kinds of PPAR subtypes might have different roles in the kidney fibrosis process. MHY-2013 blocks TGF-β-induced fibroblast activation, mainly through PPARγ activation. However, anti-inflammatory effects have through PPARβ activation. Why did you use different types of cell lines, such as NRK49F cells and NRK52E cells? And why did you use different types of stimulation, such as TGF-β and LPS? Authors need to describe clearly.
Response to reviewer: Thanks for your insightful comments. From the previous reports, it has been shown that fibroblast play most important role in the fibrosis development. The fibroblasts actively produces extracellular matrix proteins in response to fibrogenic stimuli. The most well-known fibrogenic stimuli is TGF-beta. Thus, we used fibroblasts and TGF-beta to evaluate the effect of MHY-2013. In contrast, kidney inflammation is associated with kidney epithelial damage. LPS is one of the prominent substance that induces pro-inflammatory response in the cells. Thus, we used kidney epithelial cells and LPS to evaluate anti-inflammatory effects of MHY-2013.
- Subtitle “MHY-2013 reduces LPS-induced chemokine expression in NRK49E kidney epithelial cells“ à NRK49E cells should change to NRK52E cells.
Response to reviewer: We changed NRK49E to NRK52E cells.
- NRK49F cells and NRK52E cells might have different cell behaviors. So, in figure 6, the authors need to evaluate PPARα, PPARβ, and PPARγ transcriptional activity using NRK52E cells.
Response to reviewer: Thanks for your comments. In the previous paper reported by our group, we have reported that MHY-2013 activates PPAR activity in NRK52E cells. Please refer to our previous publication.
Chung, K.W.; Ha, S.; Kim, S.M.; Kim, D.H.; An, H.J.; Lee, E.K.; Moon, H.R.; Chung, H.Y. PPARalpha/beta Activation Alleviates Age-Associated Renal Fibrosis in Sprague Dawley Rats. J Gerontol A Biol Sci Med Sci 2020, 75, 452-458, doi:10.1093/gerona/glz083.
Reviewer 3 Report
The authors demonstrated in mouse model of folic acid-induced renal fibrosis positive effect of PPAR pan- agonist on the development of fibrosis and in vitro demonstrated that the effect was mostly driven by the activation of PPAR gamma and partly PPAR beta.
I think that it is really a very good paper providing new insights into the pathogenesis of renal fibrosis. More detailed information on several aspects of the study is, however, needed.
1. Introduction is very well written and covers shortly the necessary information the reader needs to understand the topic. Some other topics should be, however, also explained (see below) including the role of PPAR alpha, beta and gamma (I appreciate figure 7, but it should also include PPAR alpha even as it is not involved in fibrosis
2. It should be, however, explained how the folic acid-induced model of renal fibrosis works and also it should be explained why the authors chose as a parameter of renal fibrosis the increased expression of those kidney damage-related genes (Havcr1, Timp2, Igfbp7, and Spp1) they report. It could be either in the Introduction (one paragraph), or in the Discussion
3. Also for the reader not experienced in the field it should be explained why they used as parameters of fibrosis vimentin and Col1A2 and also the relation between TGFbeta and SMAD is definitely not an information so well known not to be explained in the text
4. The same comment for chosen inflammatory markeres ((Tnfa, Il1b, and Ccl2 and EMr1), why these markers and how they are involved in renal fibrosis
5. Role of the expression of different genes studied in the paper could be also nicely demonstrated in the form figure (in a similar way as Figure 7)
Author Response
The authors demonstrated in mouse model of folic acid-induced renal fibrosis positive effect of PPAR pan- agonist on the development of fibrosis and in vitro demonstrated that the effect was mostly driven by the activation of PPAR gamma and partly PPAR beta.
I think that it is really a very good paper providing new insights into the pathogenesis of renal fibrosis. More detailed information on several aspects of the study is, however, needed.
Response to reviewer: Thanks for your helpful comments.
- Introduction is very well written and covers shortly the necessary information the reader needs to understand the topic. Some other topics should be, however, also explained (see below) including the role of PPAR alpha, beta and gamma (I appreciate figure 7, but it should also include PPAR alpha even as it is not involved in fibrosis
Response to reviewer: Thanks for your helpful comments. We try to explain the role of PPAR family in the discussion section with the evidence of their activity in fibrosis. To avoid the duplication, we skipped basic information in the introduction. Please understand.
- It should be, however, explained how the folic acid-induced model of renal fibrosis works and also it should be explained why the authors chose as a parameter of renal fibrosis the increased expression of those kidney damage-related genes (Havcr1, Timp2, Igfbp7, and Spp1) they report. It could be either in the Introduction (one paragraph), or in the Discussion
Response to reviewer: Thanks for your opinion. We chose the most well-known genes associated with kidney damage.
- Also for the reader not experienced in the field it should be explained why they used as parameters of fibrosis vimentin and Col1A2 and also the relation between TGFbeta and SMAD is definitely not an information so well known not to be explained in the text
Response to reviewer: Thanks for your opinion. However, the things you mentioned are basic knowledge for the fibrosis research. We do not think all the basic information should be included in the limited text.
- The same comment for chosen inflammatory markeres ((Tnfa, Il1b, and Ccl2 and EMr1), why these markers and how they are involved in renal fibrosis
Response to reviewer: Thanks for your opinion. The inflammatory markers we checked are major cytokines(Tnfa, Il1b) and chemokine (Ccl2), and marker for macrophage (Emr1). The role of these cytokines are associated with various inflammatory disease including kidney fibrosis. Due to limited space, we did not mention all information how they are related, we included general relationship between inflammation and kidney fibrosis in the discussion section.
- Role of the expression of different genes studied in the paper could be also nicely demonstrated in the form figure (in a similar way as Figure 7)
Response to reviewer: Thanks for your opinion. In the Figure 7, we summarized our findings. It would be too complicated if we includes all the gene names and their description.
Reviewer 4 Report
Minjung Son et al describes the effect of MHY-2013 in a model of renal fibrosis. They clearly show that MHY-2013 could prevent the folic acid induced interstitial fibrosis. However, I have the following remarks/questions:
1. In the introduction of the summary, the authors state that the effect of MHY-2013 in kidney fibrosis was not shown before. This is not correct since this research group already showed a beneficial effect of this compound in Age-Associated Renal Fibrosis in Sprague Dawley Rats (see reference 21)
2. From the material and methods, it is not clear how many mice in each group were used for this study. Please add this to the M&M section
3. In line with this, the authors should also mention in the figures how many mice were used for each analysis, for example from how many mice RNA was measured etc. Preferably present the data with single dots (representing a mice) in the bars of the graphs.
4. In order to investigate whether MHY-2013 inhibit the development of damage or whether it prevent fibrosis (as suggested in this MS) it would be helpful to measure for example KIM1 as a damage marker and TGF-b as a marker for induction of fibrosis. This conclusion cannot be made based on the markers in figure 1B. These additional measurements will really strengthen the conclusion. Otherwise it should be discussed as a limitation.
5. Many of figures have pictures of the (immune)histology. Except for the SR there is no quantification. How was the quantification of the Sirius Red performed. Was this done by image analysis? (this is not mentioned in the M&N, I assume ImageG?) Is it possible to use this method for the other staining’s as well? Otherwise, at least a semi quantitative scoring by two independent experienced pathologists should be done and presented in a graph.
6. The quality of the pictures I received for reviewer do not have a very good quality. Also it is preferred to show higher magnifications .
7. Several Western blots from the cell cultures are shown. However from the manuscript it is not clear how many times these cell cultures/Western blots are done. Also quantification data of the Western blots are necessary.
8. Which housekeeping gene was used for the normalization of the qPCR data?
9. It is common that the discussion start with a short summary of the results rather than a introduction of the subject as it is now.
Minor points:
In the version for review, I could not find supplemental tables 1 and 2
In the literature, MHY-2013 is mentioned as MHY2013. I think it is good to stick to that?
Author Response
Reviewer 4.
Minjung Son et al describes the effect of MHY-2013 in a model of renal fibrosis. They clearly show that MHY-2013 could prevent the folic acid induced interstitial fibrosis. However, I have the following remarks/questions:
Response to reviewer: Thanks for your helpful comments.
- In the introduction of the summary, the authors state that the effect of MHY-2013 in kidney fibrosis was not shown before. This is not correct since this research group already showed a beneficial effect of this compound in Age-Associated Renal Fibrosis in Sprague Dawley Rats (see reference 21)
Response to reviewer: Thanks for your opinion. In the last paragraph of introduction, we mentioned that we have previously showed anti-fibrotic effect in age-related kidney fibrosis model. Here, we tried to see the effect of PPAR pan agonist in the general model of kidney fibrosis. To avoid confusion, we modified our wording in the introduction section. - From the material and methods, it is not clear how many mice in each group were used for this study. Please add this to the M&M section
Response to reviewer: Thanks for your comment. We added the information in the M&M section.
- In line with this, the authors should also mention in the figures how many mice were used for each analysis, for example from how many mice RNA was measured etc. Preferably present the data with single dots (representing a mice) in the bars of the graphs.
Response to reviewer: Thanks for your comment. We added the information in the method section. We used same number for qPCR analysis in the animal experiments without exception. However, we did not change the qPCR data to single dot graphs. The graphs became too complicated with the dot expression with the multiple gene detection. (for example Figure 1B, Figure 4A)
- In order to investigate whether MHY-2013 inhibit the development of damage or whether it prevent fibrosis (as suggested in this MS) it would be helpful to measure for example KIM1 as a damage marker and TGF-b as a marker for induction of fibrosis. This conclusion cannot be made based on the markers in figure 1B. These additional measurements will really strengthen the conclusion. Otherwise it should be discussed as a limitation.
Response to reviewer: Thanks for your valuable comments. In the Figure 1B, the gene level of Havcr1 is shown, which encodes the protein KIM-1. We further checked TGF-b level according to your suggestion. The results are included in Figure 2A.
- Many of figures have pictures of the (immune)histology. Except for the SR there is no quantification. How was the quantification of the Sirius Red performed. Was this done by image analysis? (this is not mentioned in the M&N, I assume ImageG?) Is it possible to use this method for the other staining’s as well? Otherwise, at least a semi quantitative scoring by two independent experienced pathologists should be done and presented in a graph.
Response to reviewer: Thanks for your comment. We mentioned about the image calculations in the quantification and statistical analysis part in the methods. For other staining results including IHC and in situ, we did not perform all the sample, so it is not easy to quantify the whole samples done in experiments. For Sirius red staining, we quantified all the slides from the samples (n = 5~7 for each group). For this reason, we could not quantify all the histological data we performed. - The quality of the pictures I received for reviewer do not have a very good quality. Also it is preferred to show higher magnifications.
Response to reviewer: Thanks for your comment. We believe that we provided high quality resolution pictures (at least 300 dpi) at the submission stage. I do not know the quality of picture during the review process, however, we will make sure that the quality of pictures are high enough to be published.
- Several Western blots from the cell cultures are shown. However from the manuscript it is not clear how many times these cell cultures/Western blots are done. Also quantification data of the Western blots are necessary.
Response to reviewer: Thanks for your comment. We added words in the method section about the replicates of the experiments. All the western blot analysis were performed at least 3 replicates per experiment. We have checked gene expression using qPCR, which is more quantitative analysis, so we did not feel the necessity to quantify the protein expression which makes it too complex for the figure.
- Which housekeeping gene was used for the normalization of the qPCR data?
Response to reviewer: Thanks for your comment. We used Gapdh gene for the normalization. The information is shown in the qPCR graph.
- It is common that the discussion start with a short summary of the results rather than a introduction of the subject as it is now.
Response to reviewer: Thanks for your comment. We agree with the reviewer’s comment. However, there is conclusion section in the MS, we just went to discussion without short summary.
Minor points:
In the version for review, I could not find supplemental tables 1 and 2
In the literature, MHY-2013 is mentioned as MHY2013. I think it is good to stick to that?
Response to reviewer: Thanks for your comment. We will attach the supplemental tables 1 and 2. For the MHY-2013, we changed it to MHY2013.
Round 2
Reviewer 4 Report
Dear authors,
First off all, I apologise for overlooking the housekeeping gene which was already present in the figures and not recognizing HAVCR1-gene as the KIM-1.
The authors have addressed almost all my concerns, the only thing I disagree with is the quantification of the westernblots. If the authors thinks that it is not necessary the quantify them, I recommend to present the triplicates of the blots as a supplemental figure.
Author Response
Dear authors,
First off all, I apologise for overlooking the housekeeping gene which was already present in the figures and not recognizing HAVCR1-gene as the KIM-1.
The authors have addressed almost all my concerns, the only thing I disagree with is the quantification of the westernblots. If the authors thinks that it is not necessary the quantify them, I recommend to present the triplicates of the blots as a supplemental figure.
Response to reviewer: Thanks for your suggestion. According to reviewer's comments, we added densitometry analysis results for all the Western blot experiments.